# Curcumin-Rich Diet Mitigates Non-Alcoholic Fatty Liver Disease (NAFLD) by Attenuating Fat Accumulation and Improving Insulin Sensitivity in Aged Female Mice under Nutritional Stress

**DOI:** 10.3390/biology13070472

**Published:** 2024-06-26

**Authors:** Gopal Lamichhane, Da-Yeon Lee, Rienna Franks, Femi Olawale, Jong-Beom Jin, Josephine M. Egan, Yoo Kim

**Affiliations:** 1Department of Nutritional Sciences, Oklahoma State University, Stillwater, OK 74078, USA; gopal.lamichhane@okstate.edu (G.L.); dayeon.lee@okstate.edu (D.-Y.L.); rienna.franks@okstate.edu (R.F.); femi.olawale@okstate.edu (F.O.); jojin@okstate.edu (J.-B.J.); 2Laboratory of Clinical Investigation, National Institute on Aging, Baltimore, MD 21224, USA; eganj@grc.nia.nih.gov

**Keywords:** curcumin, aging, NAFLD, obesity, insulin resistance

## Abstract

**Simple Summary:**

This study aimed to understand how dietary curcumin influences metabolic abnormalities induced by a high-fat high-sugar diet (HFHSD) in aged female mice. We observed that curcumin effectively reduced body fat accumulation, steatosis of liver, and insulin resistance caused by a nutritional challenge, suggesting its potential in mitigating age-related metabolic disturbances. These findings highlight curcumin supplementation as a potential strategy to address metabolic issues associated with female aging.

**Abstract:**

Background: The high incidence of metabolic syndrome in the elderly poses a significant challenge to the healthcare system, emphasizing the need for interventions tailored to geriatric patients. Given the limited focus on females in previous studies, this research aimed to evaluate the effects of dietary curcumin on obesity and NAFLD outcomes in naturally aged (18-month-old) female mice. Methods: Female C57BL/6 mice aged 18 months were fed a normal chow diet (NCD) and a HFHSD, with or without curcumin (0.4% *w*/*w*), for an 8-week period. Parameters included food intake, body weight, insulin tolerance test (ITT), glucose tolerance test (GTT), percentage fat mass, hepatic triglyceride, and cholesterol levels, and a histological examination for NAFLD detection, qPCR, and immunoblotting analyses were performed. Results: The cumulative body weight gain after 8 weeks in the aged female mice supplemented with curcumin and fed an HFHSD was significantly lower (10.84 ± 1.09 g) compared to those fed a HFHSD alone (15.28 ± 1.26 g). Curcumin supplementation also resulted in reduced total body fat (HFHSD group 50.83 ± 1.71% vs. HFHSD+CUR 41.46 ± 3.21%), decreased epidydimal fat mass (HFHSD: 3.79 ± 0.29 g vs. HFHSD+CUR: 2.66 ± 0.30 g), and repaired adipogenic signaling in the white adipose tissue. Furthermore, curcumin lowered triglyceride and cholesterol deposition in the liver, preventing hepatic steatosis and improving hepatic insulin sensitivity. Conclusions: Curcumin demonstrates the ability to ameliorate the deleterious effects of HFHSD in aged female mice by reducing body fat composition, modulating adipogenic signaling in the white adipose tissue, and improving insulin homeostasis and non-alcoholic fatty deposition in the liver.

## 1. Introduction

The demographic shift toward an aging population and the consequent increase in age-associated diseases is a significant challenge to social and healthcare infrastructures worldwide [1]. Changes in body composition with age, including increased susceptibility to abdominal obesity, contribute to insulin resistance and metabolic syndrome, making geriatric obesity of particularly concern [2,3]. Factors such as a sedentary lifestyle post-retirement, altered metabolism, and the consumption of high-energy-density foods cause an imbalance between energy intake and expenditure, resulting in excess fat storage and obesity [4,5]. Obesity, in turn, exacerbates the risk of various diseases, including diabetes, cardiovascular diseases, dementia, non-alcoholic fatty liver disease (NAFLD), and cancer [6].

The prevalence of NAFLD, a hepatic abnormality resulting from non-alcoholic fatty deposition and steatohepatitis, has increased together with a rise in metabolic syndrome, leading to conditions like hepatic cirrhosis and cancer [7]. Approximately 30 percent of the US population is affected by NAFLD, and this number is expected to grow further, potentially becoming a leading cause of hepatic transplantation, surpassing hepatitis C [8]. The risk of NAFLD is higher in the elderly due to age-related predispositions to hyperlipidemia, high blood pressure, diabetes, and obesity [9]. Additionally, it is noteworthy that NAFLD prevalence is higher in men compared to women [10], with variations observed post-menopause [11,12,13,14,15]. Challenges in addressing NAFLD include the absence of effective biomarkers that can identify the severity of steatohepatitis [16,17]. This leaves histological examination, an invasive technique with high costs and potential complications, as the gold standard for diagnosis, which cannot be considered for all patients [16,17]. Recent techniques such as advanced metabolomics and lipidomics have shown promising potential to be developed into non-invasive identification tools for NAFLD, given that they are further validated [16,18]. Furthermore, there are limited effective interventions available, with lifestyle modifications, bariatric surgery, insulin sensitizers, and other weight management strategies being the primary approaches [8,13]. More research is ongoing to explore the possible use of PPAR ligands, miRNA, glucagon-like peptide (GLP) analogues, farnesoid X receptor (FXR) agonists, chemokine receptor (CCR)2/CCR5 antagonists, probiotics, antibiotics, vitamin E, and statins in the management of NAFLD [19].

Metabolic pathways in the body depend on the feeding status of the organism. During the fed state, complex food is broken down into simpler forms and absorbed into the body. Postprandial increases in blood glucose cause the release of insulin from the pancreas, which activates downstream signaling for adipogenesis, protein synthesis, and glycogenesis. The activation of these anabolic signaling cascades causes excess food to be stored as glycogen and fat for future use. However, during fasting, blood glucose levels are low, causing a lower secretion of insulin and an increased secretion of glucagon, which converts the stored glycogen into glucose, maintaining glucose homeostasis. Fasting also increases fatty acid oxidation and gluconeogenesis to provide energy to the body [20]. During insulin resistance, this intricate balance of pathways is disturbed. As a result, fat tissue releases more fatty acids into the circulation as they become insensitive to insulin, which in turn further worsens insulin resistance by altering insulin signaling in other organs [21].

Insulin resistance-mediated type 2 diabetes is a major challenge in aging, exacerbated by abdominal obesity [22,23]. Excessive fat deposition in the adipose tissue during obesity leads to adipocyte hypertrophy, inflammation, hypoxia, and macrophage recruitment, ultimately resulting in the release of free fatty acids, reactive oxygen species, and proinflammatory cytokines. These released fats and lipids then accumulate in non-adipose organs such as the liver and muscles, leading to ectopic obesity and cellular organelle damage, including lysosomes, mitochondria, and endoplasmic reticulum, further increasing oxidative stress and inflammation. This low-grade systemic inflammation disrupts insulin signaling pathway [24,25]. Therefore, controlling body weight is an effective strategy for preventing NAFLD [26] and type 2 diabetes [27].

Research has demonstrated the potential role of food bioactive compounds in managing obesity and related disorders. Food-derived bioactive compounds, such as curcumin, resveratrol, tea polyphenols, anthocyanins, and bioactive compounds from coffee, have shown improvements in fatty liver disease by reducing oxidative stress, steatosis, and inflammation in previous studies [28]. Curcumin, derived from turmeric, is a bioactive polyphenol that has been used in indigenous medicine in India to treat various diseases [29]. Since its isolation in pure form, research has shown its efficiency in managing cardiovascular and metabolic diseases, acting as an antioxidant and anti-inflammatory agent [30]. We have also demonstrated the role of curcumin in the management of Alzheimer’s disease, hepatic cellular senescence, and insulin homeostasis in mice of different ages under dietary challenges [31,32,33].

Despite the physiological differences between genders that can alter obesity outcomes, most preclinical research is carried out in male animals [34,35]. Factors such as higher fat mass in females, discrepancies due to different sex hormones, and behavioral disparities are among the gender-specific factors affecting obesity that are often overlooked [36]. The current evidence does not clearly elucidate how curcumin affects obesity and NAFLD outcomes in naturally aged female mice challenged with an HFHSD. Thus, this research aims to evaluate the effect of curcumin on HFHSD-challenged metabolic health and obesity outcomes in naturally aged female mice.

## 2. Materials and Methods

### 2.1. Animals and Treatment

Eighteen-month-old-aged female C57BL/6 mice were obtained from the National Institute of Aging, housed at Charles River Laboratories (Raleigh, NC, USA), and acclimatized for a week at the Animal Care Facilities at Oklahoma State University by providing ad libitum access to a standard chow diet and water. Baseline body weight, 6 h fasting, and fed blood glucose levels were measured for all mice at the end of acclimation period. The mice were then randomly divided into four dietary groups (*n* = 7–9): normal chow diet (NCD), curcumin-supplemented (4 g/kg) normal chow diet (NCD+CUR), high-fat high-sugar diet (HFHSD), and curcumin-supplemented (4 g/kg) high-fat high-sugar diet (HFHSD+CUR) fed groups. Ad libitum access to a customized diet (purchased from Dyets Inc., Bethlehem, PA, USA) and water was provided throughout the 8-week study period with the weekly monitoring of food intake and body weight. Mice were sacrificed following a 24 h fasting or 24 h fasting followed by a 3 h refeeding period (for all group), and metabolic organs like the liver and white adipose tissue were harvested for future analysis.

### 2.2. MRI Imaging

The MRI imaging of mice was conducted using the M3 Compact MRI System from Aspect Imaging Ltd. (Scintica Inc., Webster, TX, USA). Mice were kept in a chamber continuously supplied with a mixture of air and isoflurane until anesthetized. Immobilized mice were positioned in an MRI coil with a continuous supply of isoflurane-mixed air for anesthesia. Mice were then scanned using T1 and Dixon mode to access fat deposition in the body. Images from T1 were scanned using ImageJ to quantify the relative fat mass in mice.

### 2.3. Body Composition Analysis Using PIXIMUS

A day before the termination of the study, the mice were anesthetized by ketamine injection. Once immobilized, mice were fixed in a specimen tray and scanned using LUNAR PIXI from GR LUNAR Corporate (Madison, WI, USA). After the completion of the scan, mice were kept warm at 37 °C for recovery. The head of the mice was excluded from data analysis. Bone mineral density (BMD), bone mineral content (BMC), lean weight, fat weight, and percentage fat were recorded and analyzed.

### 2.4. Glucose and Insulin Tolerance Test

In the sixth week after the intervention, mice were fasted for a 16 h period before the glucose tolerance test (GTT). They were then given 2 g/kg BW glucose (Alpha Teknova Inc., Hollister, CA, USA) via intraperitoneal injection. Blood glucose was measured at 0 min, 15, 30, 60, 90, and 120 min post-injection from the caudal vein using the Contour^®^ Next EZ glucometer (Parsippany, NJ, USA).

For insulin tolerance test (ITT), mice were injected with 0.40 IU/kg intraperitoneal insulin (Novolin R from Novo Nordisk, Bagsvaerd, Denmark) following 6 h of fasting. Blood glucose levels were then measured at 0 min, 15, 30, 60, 90, and 120 min post-injection.

### 2.5. Immunoblotting Analysis

Portions of hepatic tissue (40–50 mg) and epidydimal white adipose tissue stored at −80 °C were homogenized in T-PER™ Tissue Protein Extraction Reagent (ref 78510) from Thermo Scientific (Rockford, IL, USA), premixed with one tablet each/10 mL of cOmplete Mini protease inhibitor cocktail (ref 11836170001) and PhosSTOP phosphatase inhibitor (ref 04906837001) from Roche (Manheim, Germany) using the OMNI BeadRuptor 24 (Omni-Inc., Kennesaw, GA, USA). Following protein quantification using the BCA assay kit (Thermo Scientific, Rockford, IL, USA), an equal amount of protein sample was resolved in SDS-PAGE. Proteins in the gel were transferred to polyvinylidene fluoride (PVDF) membranes, blocked for an hour using a blocking reagent (LI-COR, Lincoln, NE, USA) at room temperature. The blocked membranes were then incubated overnight with primary antibodies: Phospho-Akt (Ser473) (D9E) XP^®^ (cat no. 4060), Akt (pan) (C67E7) (cat no. 4691), phospho-mTOR (Ser2448) (cat no. 2971), mTOR (cat no. 2972), phospho-PRAS40 (Thr246) (D4D2) XP^®^ (cat no. 13175), PRAS40 (D23C7) XP^®^ (cat no. 2691), FoxO1 (C29H4) (cat no. 2880), phospho-acetyl-CoA carboxylase (p-ACC, cat no. 3661), ACC (cat no. 3676), phosphor-AMP-activated protein kinase α (p-AMPKα, cat no. 5831), AMPKα (D5A2) (cat. no. 2532), glyceraldehyde 3-phosphate dehydrogenase (GAPDH (D16H11), cat no. 5174), and β-tubulin (cat no. 2146) from Cell Signaling Technology (Denver, MA, USA). Lipoprotein lipase (LPL, cat no. 16899-1-AP) was purchased from ProteinTech (Rosemont, IL, USA). On the next day, the primary antibodies were removed, and the membranes were washed three times using tris-based saline with Tween 20 (TBS-T) and incubated for 1 h with anti-rabbit secondary antibodies (cat no. 7074) from Cell Signaling Technology dissolved in TBS-T with 5% skimmed milk. The secondary antibody solution was then removed, and the membranes were washed three times using TBS-T. The membranes were than developed using chemiluminescent substrate (ref 34578) from Thermo Scientific (Rockford, IL, USA) and visualized on Green X-Ray Film (cat no. 1148B77) from Thomas Scientific (Swedesboro, NJ, USA). The membranes were scanned and quantified using the ImageJ software (v. 1.54d, NIH, Bethesda, MA, USA), normalized with their respective β-tubulin or GAPDH, followed by normalization with phosphorylated proteins to their total forms.

### 2.6. Hepatic Triglyceride and Cholesterol Measurements

The method explained by Peng et al. was used for the total lipid extraction from the liver tissue [37]. Briefly, around 75 mg of liver was weighed and homogenized by adding 1 mL of a 50 mM NaCl solution using a Dounce homogenizer. The homogenate thus obtained was mixed with 5 mL of a 2:1 chloroform: methanol mixture to extract the total lipids. After vortexing (15 s for 3 times), the tubes were centrifuged at 1000 rpm for 10 min. The upper aqueous phase was discarded, and 1 mL of 50 mM NaCl was carefully added without disturbing the interphase. The aqueous upper layer above the interphase was discarded again, followed by the addition of 1 mL methanol. The total volume was adjusted to 6 mL for all tubes by adding a 2:1 mixture of chloroform: methanol, thoroughly mixed by vertexing, and centrifuged at 1000 rpm for 10 min to separate the upper lipid extract and cellular debris as a pellet at the bottom. A total of 300 µL (for cholesterol) and 75 µL (for triglyceride) of the lipid extract was transferred to a new separate tube, followed by the addition of 75 µL and 18.75 µL of 10% triton X100 (in acetone), respectively. The tubes were left overnight in a fume hood to dry by the evaporation of solvent. Total cholesterol and triglyceride were determined using the Pointe Cholesterol (liquid) Reagent Set and Pointe Triglyceride (GPO) (Liquid) Reagent Set from MedTest Dx (Canton, MI, USA). The final cholesterol and triglyceride levels were expressed in mg cholesterol/triglyceride per gram of liver weight.

### 2.7. Histological Analysis

The hepatic tissue was fixed in 10% formaldehyde, dehydrated, and embedded in a paraffin block. A 5 µm slice was prepared using a microtome and dehydrated overnight. This was followed by staining using the Trichrome, Masson, Aniline Blue Stain Kit supplied by Newcomersupply (catalog no. 9179B, Middleton, WI, USA), following the manufacturer’s protocol. Stained sections were then imaged at 10× total magnification using the Keyence BZ-X710 All-in-one Fluorescence Microscope (Itasca, IL, USA). Images were anonymously graded for fatty liver in the range of 0–3: 0 corresponds to normal liver, 1 to mild fatty liver, 2 to moderate fatty liver, and 3 to severe fatty liver, and the overall ratings for the groups were analyzed for significant differences [38].

### 2.8. Real-Time Quantitative Polymerase Chain Reaction (qPCR)

The total RNA was isolated using the TRIzol Reagent (Thermo Fisher Scientific, Pleasanton, CA, USA), and cDNA was synthesized using the iScript™ cDNA Synthesis Kit (Bio-Rad Laboratories, Inc., Hercules, CA, USA). The expression of the genes was assessed using the SYBR^®^ Green PCR Master Mix (Applied Biosystems; Thermo Fisher Scientific, Waltham, MA, USA) on a CFX Opus 384 Real-Time PCR System (Bio-Rad Laboratories, Hercules, CA, USA). Mouse primers for acetyl-CoA carboxylase 1 (ACC1), ACC2, peroxisome proliferator-activated receptor α (PPARα), PPARγ, carbohydrate-responsive element-binding protein (ChREBP), sterol regulatory element-binding protein (SREBP), and lipoprotein lipase (LPL) were used as the template. The primer sequences are provided in Appendix A. The data were normalized to 18S ribosomal RNA and β-actin.

### 2.9. Statistical Analysis

All data were analyzed using GraphPad Prism (V10.2.2: GraphPad Inc., Boston, MA, USA). A two-way ANOVA followed by Tukey’s multiple comparison test was used for body weight, ITT, GTT, and FER. An unpaired *t*-test was used for analyzing the AUC of ITT and GTT, hepatic cholesterol, and triglyceride contents, percentage fat, epidydimal adipose tissue weight, qPCR, and histology grading results. Spearman correlation analysis was conducted between body weight and hepatic triglyceride and cholesterol levels. All data were presented as mean ± SEM, and statistical significance was checked at * *p* < 0.05, ** *p* < 0.01, and *** *p* < 0.001 levels.

## 3. Results

### 3.1. Curcumin Reduced Body Fat Mass in HFHSD-Fed Aged Female Mice

Body weight gain after 8 weeks of dietary intervention is presented in Figure 1A. While the NCD-fed groups maintained their body weight throughout the experiment period, the HFHSD-fed mice group gained body weight continuously, reaching the saturation level (14.69 ± 0.85 g) at week 6. The body weight gain after 6 weeks of the curcumin-supplemented mice was 10.27 ± 0.95 g, significantly lower than that of the HFHSD-fed group. A significant difference in body weight gain between those groups was seen starting from week 5 and maintained until the end of study. The difference in the body weight gain was not due to differences in diet intake, as there was similar accumulated food intake among all the treatment groups (Figure 1B). However, the addition of curcumin to the diet significantly reduced the food efficiency ratio (FER), as seen in Figure 1C.

Guided by a significant difference in body weight gain for HFHSD-fed mice, we were curious to see if there was a change in the body composition of those mice. The PIXIMUS scan of mice showed a significantly higher percentage of fat tissue in HFHSD-fed mice (50.83 ± 1.71%) compared to HFHSD+CUR-fed mice (41.46 ± 3.21%) (Figure 1E), which was further confirmed by the T1 MRI scan image (Figure 1D), and its quantification using ImageJ showed a greater white area. These results are further supported by a significantly high mass of epididymal white adipose tissue (WAT, HFHSD: 3.79 ± 0.29 g and HFHSD+CUR: 2.66 ± 0.30 g) in HFHSD-fed mice compared to HFHSD+CUR-fed mice. We did not observe any difference in lean body mass or bone density between the groups during the PIXIMUS scan (Figure 1H–J).

### 3.2. Curcumin Modulated Hepatic Insulin Signaling in HFHSD-Fed Female Mice

In accordance with increased body weight and fat mass, we observed significant differences in insulin sensitivity in the HFHSD+CUR group compared to HFHSD-fed mice during the ITT assay. The mean glucose level was markedly lower at the 60- and 90-min time points (*p* < 0.063 and *p* < 0.01, respectively). This was further supported by the area under the curve (AUC), which was significantly lower for the curcumin-supplemented group compared to the control (Figure 2A). However, we did not observe any change in glucose tolerance by curcumin supplementation (Figure 2B).

Following this observation, we were curious as to whether the increased insulin sensitivity in the ITT assay was reflected in the insulin signaling pathway. Evidently, p-Akt at Ser473 was highly expressed only during refeeding in HFHSD+CUR-fed mice. In HFHSD-fed mice, there was more activated insulin signaling even during the prolonged fasting condition. The higher expression of p-Akt at Ser473 in HFHSD-fed fasted mice compared to HFHSD + CUR-fed fasted mice suggested insulin resistance in the HFHSD group compared to the curcumin-fed counterpart. The result is further supported by downstream markers, including p-mTOR and p-PRAS40, which show the same trends in the refed mice of the two groups. Another downstream marker and gluconeogenesis regulation marker, forkhead box protein O1 (FoxO1), was highly expressed during the fasted condition in the curcumin-supplemented group, which was diminished in the refed condition. However, in the HFHSD group, the FoxO1 expression was lower during fasting and did not turn off even in the refed condition.

### 3.3. Curcumin Ameliorated Triglyceride and Cholesterol Deposition in the Liver of HFHSD-Fed Mice

As the liver is the primary metabolic organ in the body, and we observed different insulin sensitivities between HFHSD- and HFHSD+CUR-fed mice, we then decided to evaluate if it has any influence on hepatic lipid homeostasis and steatosis in the liver of mice. The quantification of triglyceride and cholesterol in the liver homogenate of mice showed significantly higher triglyceride deposition in the liver of fasted (89.53 ± 1.68 mg/g tissue weight) and refed mice (80.50 ± 8.09 mg/g tissue weight) in the HFHSD-fed group compared to the HFHSD+CUR group (Figure 3A). In the HFHSD+CUR group, the triglyceride levels were observed to be 65.41 ± 14.55 mg/g tissue weight in the fasted mice group and 53.0 ± 11.22 mg/g tissue weight in the refed mice group. Also, the cholesterol contents were significantly lower among the refed groups (1.81 ± 0.095 mg/g tissue weight in the HFHSD-fed group vs. 1.38 ± 0.19 mg/g tissue weight in the HFHSD+CUR-fed group) (Figure 3B). Furthermore, we observed a significantly (*p* < 0.05) positive correlation between total body weight gain and hepatic triglyceride (correlation coefficient of 0.68) and body weight gain and cholesterol (correlation coefficient of 0.59) deposition, indicating the vulnerability of mice with a higher body weight to develop fatty liver disease (Figure 3C).

### 3.4. Curcumin Ameliorated HFHSD-Induced Hepatic Steatosis in Mice

The accumulation of lipid in hepatocytes is considered a hallmark of non-alcoholic fatty liver disease (NAFLD) induction and progression. Hepatic steatosis was exacerbated in HFHSD feeding. The overall grading for the steatosis was calculated to be exceedingly high at 2.71 ± 0.16 in the HFHSD-fed group, while that for the HFHSD+CUR group was 1.57 ± 0.20 (Figure 3D,E). Mild age-associated steatosis was evident in NCD- and NCD+CUR-fed mice, represented by small fat droplets in histology sections and a grading of 1.3 ± 0.38 and 1.3 ± 0.19 in NCD and NCD+CUR, respectively. Overall, a high degree of micro- and macrosteatosis was observed in the HFHSD-fed mice group, which was ameliorated by curcumin supplementation in their diet.

### 3.5. Curcumin Downregulated Adipogenesis- and Lipogenesis-Related Genes in white Adipose Tissues of Mice

We observed a 4-fold decrease in the acetyl coenzyme carboxylase 1 (ACC1) gene, a rate-limiting enzyme for fatty acid synthesis, in the curcumin-supplemented group compared to the control [39]. Additionally, we observed the downregulation of the carbohydrate response element-binding protein (ChREBP), a gene that increases the likelihood of NAFLD [40]; ACC2, a gene playing a role in fatty acid synthesis [41]; AND peroxisome proliferator-activated receptor gamma (PPARγ), an adipogenic gene [42]. Meanwhile, the increased expression of the lipolytic gene, LPL [43], was evident in HFHSD+CUR-fed mice compared to the HFHSD control (Figure 4A). These results are further confirmed by the upregulated ratio of p-ACC1/ACC1, ratio of p-AMPKα/AMPKα, and LPL protein expression levels (Figure 4B). Thus, reduced body weight, hepatic triglyceride, and epididymal fat deposition in these mice might be the result of down/upregulated genes in the curcumin-treated group.

## 4. Discussion

Body weight gain, marked by increased intra-abdominal fat, raises the risk of metabolic syndrome, which in turn increases the risk of metabolic and cardiovascular diseases [44]. A Western-style diet high in fat and sugar is considered a major contributor to weight gain and these diseases [45]. We observed a significantly higher body weight gain in HFHSD-fed aged female mice starting from the fifth week until the end of the study period, which is in accordance with other studies on mice of different ages and sexes [32,33]. This implies that curcumin can lower HFHSD-induced body weight gain irrespective of age and sex. However, curcumin supplementation had no impact on body weight gain for NCD-fed mice. Thus, curcumin supplementation is more likely to be effective in reducing body weight gain under nutritionally challenged circumstance. Curcumin also reduced the food efficiency ratio (FER) of HFHSD without any difference in the total food intake, highlighting the effectiveness of curcumin in lowering HFHSD-induced body weight gain.

Insulin resistance in obesity is one of the key risk factors causing obesity-induced type 2 diabetes [25]. With aging, the risk of insulin resistance increases in females due to menopause-associated decreases in estrogen, which has a protective effect [46]. Obesity-mediated low-grade inflammation, the expression of adipose tissue-specific cytokines, and inflammatory cytokines secreted by visceral adipocyte are some of the causes of systemic insulin resistance leading to low insulin sensitivity [25]. Age-associated muscular dysfunction, abdominal obesity, and resulting insulin resistance are considered major risk factors that increase the incidence of type 2 diabetes [23,47,48], and this risk can be lowered by reducing obesity [49,50]. As the liver plays a key role in maintaining glucose homeostasis, maintaining hepatic insulin sensitivity is considered even more important for managing diabetes [51]. Interestingly, we observed an ameliorative effect of curcumin in HFHSD-induced insulin insensitivity in HFHSD-fed obese aged mice (Figure 2A), which was further supported by downstream markers of the insulin signaling pathway in the liver (Figure 2C,D). We observed a lower activation of downstream insulin signaling molecules in fasted mice in the HFHSD+CUR group, as represented by low levels of p-Akt at Ser473, p-mTOR, and p-PRAS40, compared to the HFHSD-fed controls. This signifies that there was less insulin signal transduction in those mice during fasting than in HFHSD-fasted mice. A similar suppression of Akt-mTOR signaling was also observed in a previous study in rats fed a high-fat diet supplemented with curcumin [52]. Lower blood glucose during fasting condition diminishes insulin secretion from pancreatic β-cells while releasing glucagon, which activates PPARα and PGC-1α, increasing fatty acid oxidation, gluconeogenesis, and ketogenesis [53]. Transcription factors for lipogenesis, adipogenesis, and cholesterol synthesis are also inhibited during this phase [53]. The downregulated insulin signaling in HFHSD+CUR-fed mice might result from the ameliorative effect of curcumin on HFHSD-induced hyperinsulinemia and the resulting signaling [32]. Conversely, during refeeding, key insulin signaling nodules were more activated in the curcumin-supplemented group than in the HFHSD control group. This indicates a higher insulin sensitivity in the curcumin-supplemented group than in HFHSD control mice, similar to a previous study in 5-week-old mice [54]. The results are also supported by the increased expression of FoxO1, a transcription factor that drives gluconeogenesis [55], in the HFHSD+CUR group during fasting conditions, which was turned off by Akt activation during refeeding. This phenomenon was less distinct in the HFHSD-fed counterparts. Overall, the disrupted insulin signaling due to aging coupled with dietary HFHSD challenge in mice was ameliorated by curcumin supplementation. However, we did not observe any difference in glucose disposal efficiency in the HFHSD+CUR group compared to the HFHSD group. The reason for this is possibly due to the faster degradation of insulin in the HFHSD+CUR-supplemented group, as curcumin can increase the expression of hepatic insulin-degrading enzymes [56].

Metabolic dysfunction in muscle, liver, and white adipose tissue are considered major alterations during dietary challenges with HFHSD [57]. Altered mitochondrial function and dysregulated nutrient sensing are even considered hallmarks of aging [58]. With increased obesity, adipocytes become larger in size to store excess fat and eventually become dysfunctional. These dysfunctional adipocytes recruit macrophages, increase inflammation, cause insulin resistance, and release excess free fatty acids into the bloodstream. These released free fatty acids are stored in non-adipose organs like the liver and muscle, causing lipotoxicity [24]. In this study, we used naturally aged female mice fed with HFHSD to simulate the cumulative impact of aging and a Western diet. We observed that a high-fat high-sugar diet significantly increased adipose tissue mass, as observed in MRI and PIXIMUS scans and a higher amount of epidydimal fat mass. This fat mass was ameliorated by curcumin supplementation via downregulating adipogenesis- and fatty acid synthesis-related genes in mice (Figure 4). The expression of ACC1, a gene responsible for synthesizing enzymes that catalyze the rate-limiting steps in fatty acid synthesis [59], ChREBP, an inducer of fatty acid synthesis elongation and desaturation [60], and an adipogenic gene, PPARγ [61], were significantly downregulated in curcumin-refed mice. Moreover, the lipolytic gene LPL [43] was significantly upregulated. These results are further confirmed at the translational level using immunoblotting.

The liver is the primary metabolic organ in the body, and obesity causes an increased deposition of fat in the liver, causing NAFLD [62,63]. Obesity plays a role in both the initiation and progression of NAFLD by causing concomitant insulin resistance, type 2 diabetes, metabolic syndrome, inflammation, and DNA methylation [64]. In our study, we also found that body weight gain had a significant moderate positive correlation with the total triglycerides and cholesterol deposited in liver (Figure 3C). This might be due to the obesity-induced hypertrophy of adipocytes, which release free fatty acids causing ectopic fat deposition [24]. The higher hepatic triglycerides and cholesterol in HFHSD-fed mice were significantly reduced by curcumin supplementation (Figure 3A,B). This finding is further supported by the histological analysis, in which HFHSD-fed mice showed the highest macrosteatosis and microsteatosis resulting from lipid deposition, which was lower in the curcumin-supplemented group (Figure 3D). The overall grading score for steatosis was also significantly lower in curcumin-supplemented HFHSD-fed mice compared to the HFHSD control (Figure 3E). This indicates that hepatic steatosis can be ameliorated by curcumin supplementation. A similar improvement in hepatic steatosis was observed by Li et al. in 6-week-old male mice supplemented with curcumin [65]. Improving hepatic steatosis and triglycerides indeed improves insulin sensitivity in the liver [66]. 

While we have demonstrated a significant improvement in hepatic steatosis, reduced fat mass, and improved adipogenic signaling in the curcumin-supplemented group, we had some limitations. Although we had previously shown that curcumin improves insulin homeostasis by preserving pancreatic islet integrity [56], evaluating pancreatic health in aged female mice would provide better insights into insulin resistance and improved insulin sensitivity, especially in aged models of curcumin-treated mice. Furthermore, conducting research in aged female human participant might provide better insights into how curcumin responds in clinical settings.

## 5. Conclusions

In conclusion, our study provides compelling evidence for the beneficial effects of curcumin supplementation in mitigating HFHSD-induced body weight gain and NAFLD in naturally aged female mice. This ameliorative potential was due to improvements in impaired adipogenic signaling in WAT, reduced overall body fat, diminished hepatic triglyceride deposition, and enhanced hepatic insulin sensitivity in mice. Further research to explore the effects of curcumin supplementation in geriatric participants can provide better clinical insights into its response in humans.

## Figures and Tables

**Figure 1 biology-13-00472-f001:**
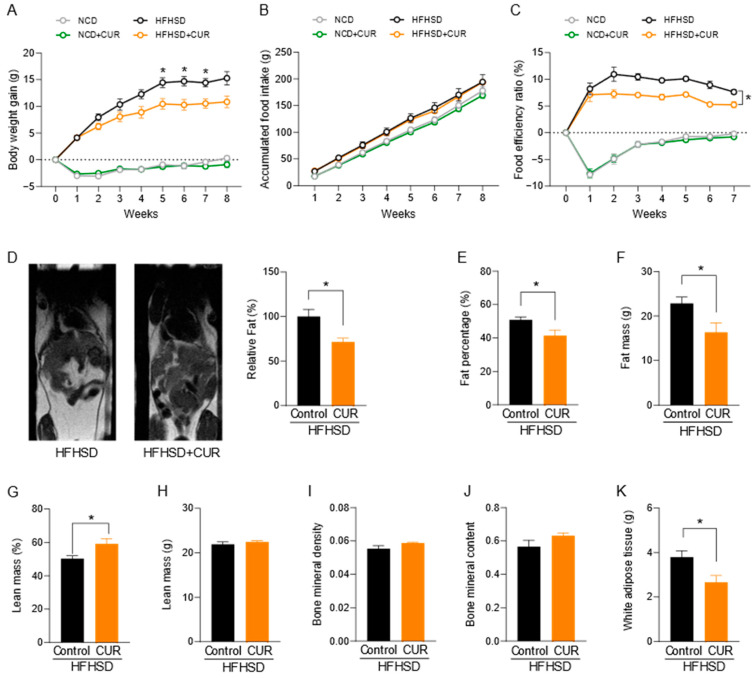
(**A**) Body weight gain (*n* = 8–9 per group), (**B**) accumulated food intake, and (**C**) food efficiency ratio (FER) for mice fed with normal chow diet (NCD) and high-fat high-sugar diet (HFHSD) with and without curcumin (CUR). (**D**) MRI image (T1 scan) and quantification of MRI image for fat (*n* = 3 per group); (**E**–**J**) percentage fat mass, absolute fat mass, percentage lean mass, absolute lean mass, bone mineral density (BMD), and bone mineral content (BMC) calculated using a PIXIMUS scan (*n* = 5–6 per group); and (**K**) epididymal WAT mass (**G**) of HFHSD- and HFHSD+CUR-fed female mice. A two-way repeated measures mixed ANOVA was used to analyze the body weight gain, accumulated food intake, and FER. An unpaired Student’s *t*-test was used to analyze MRI scan, PIXIMUS scan, and epididymal fat deposit. Data are represented as mean ± SEM and significant differences between the groups are presented as * *p* < 0.05 vs. control.

**Figure 2 biology-13-00472-f002:**
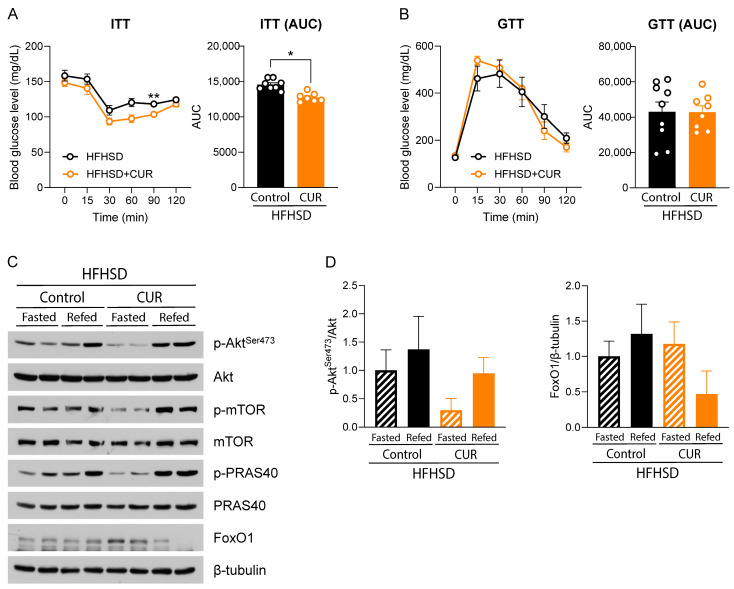
(**A**) ITT at week 8 and (**B**) GTT at week 7 in aged female mice (*n* = 8–9). HFHSD-fed mice with and without curcumin (CUR) were fasted 24 h and refed 3 h ad libitum. (**C**) Immunoblots of total liver lysates and (**D**) quantification for expressions of p-Akt (Ser473), Akt, p-mTOR, mTOR, p-PRAS40, PRAS40, and FoxO1. A two-way repeated measures mixed ANOVA was used to analyze ITT and GTT, and a one-way ANOVA was used for immunoblot quantifications. An unpaired Student’s *t*-test was used to analyze AUCs. Results are presented as mean ± SEM and significant differences are presented as * *p* < 0.05 and ** *p* < 0.01.

**Figure 3 biology-13-00472-f003:**
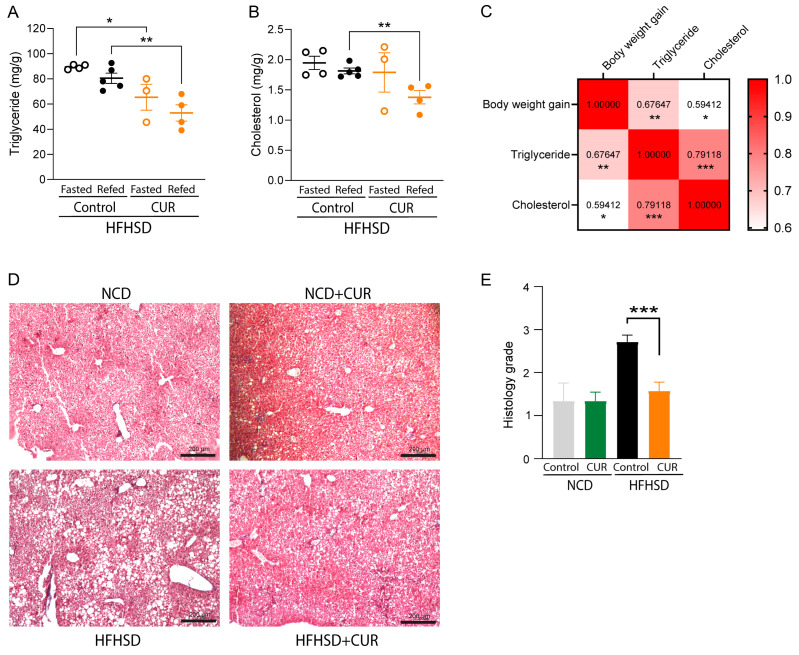
(**A**) Hepatic triglycerides, (**B**) cholesterol levels (*n* = 3–5), and their (**C**) Spearman correlation coefficient with body weight gain. Higher value indicates a strong positive correlation between the two variables, and the significance level, indicated by *, **, and ***, represent whether the relation observed was significant. (**D**) Trichome staining and (**E**) histological grading of hepatic tissue to evaluate the level of hepatic steatosis. Images were anonymously graded for fatty liver in the range of 0–3: 0 corresponds to normal liver, 1 to mild fatty liver, 2 to moderate fatty liver, and 3 to severe fatty liver. An unpaired Student’s *t*-test was used to analyze grading for steatosis. * *p* < 0.05, ** *p* < 0.01, and *** *p* < 0.001.

**Figure 4 biology-13-00472-f004:**
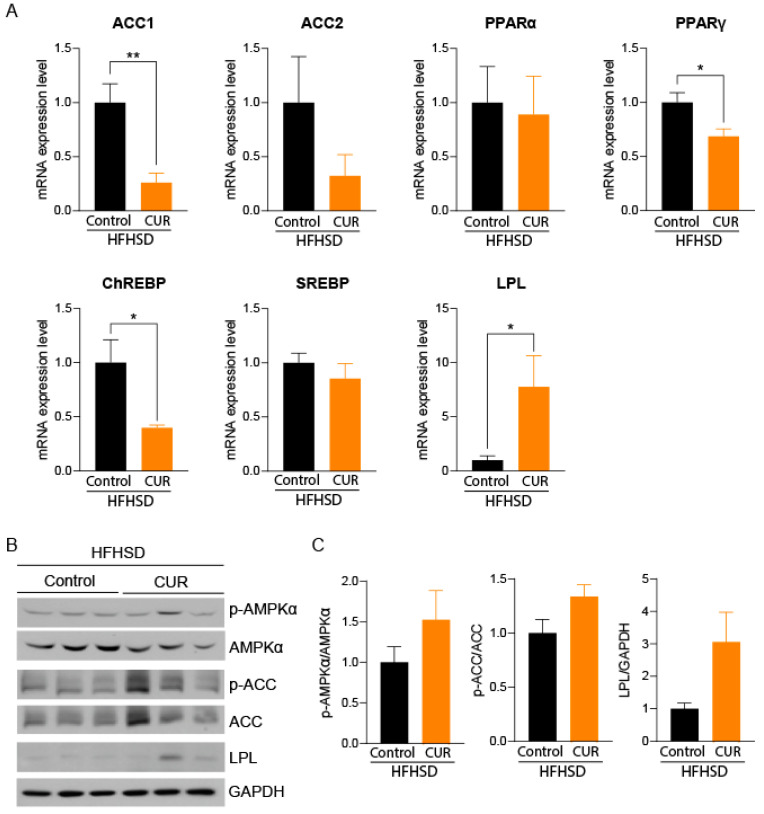
(**A**) qPCR of epididymal white adipose tissue in refed female mice (*n* = 3–4) and (**B**) immunoblotting analysis of white adipose tissue and (**C**) their quantification. Gene expression was normalized with the average respective β-actin and 18S ribosomal RNA expression levels. Protein expression was calculated as the ratio of p-AMPKα/AMPKα and p-ACC/ACC, and LPL was normalized with GAPDH expression levels. Expressed levels relative to the HFHSD refeeding group were used to investigate the effect of curcumin on gene or protein expression in refed mice. Results are expressed as mean ± SEM, and significant differences are presented as * *p* < 0.05 and ** *p* < 0.01 vs. the respective HFHSD control.

## Data Availability

The datasets used and/or analyzed during the current study are available from the corresponding author upon reasonable request.

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
