# Peer review of "Curcumin-Rich Diet Mitigates Non-Alcoholic Fatty Liver Disease (NAFLD) by Attenuating Fat Accumulation and Improving Insulin Sensitivity in Aged Female Mice under Nutritional Stress"

_biology, 2024, doi:10.3390/biology13070472_

Round 1

Reviewer 1 Report

Comments and Suggestions for Authors

this is a generally well written and structured article, with very interesting results.

it is necessary, however, to reformat the graphs where the p values appear (it is sufficient the *) and the p value (significance) information of the results must be imbedded in the text

Author Response

We appreciate that the reviewer recognize the significance of our study and provided us with helpful comments. We have carefully addressed all the points raised in our revised manuscript using the blue color. In addition, the authors revised sentences regarding grammatical errors throughout the manuscript.

Reviewer(s)' Comments to Author:

Reviewer 1:

  1. It is necessary, however, to reformat the graphs where the p values appear (it is sufficient the *) and the p value (significance) information of the results must be imbedded in the text.

Thank you for the comment. We have modified graphs displaying p-values and the p-value information of the results in the text (lines 268-269). 

Reviewer 2 Report

Comments and Suggestions for Authors

The article “Curcumin-Rich Diet Mitigates Non-Alcoholic Fatty Liver Disease (NAFLD) by Attenuating Fat accumulation and Improving Insulin Sensitivity in Aged Female Mice Under Nutritional Stress” is an interesting work, but major information is missing to conclude.

The article is well-organized and presents promising results.

The author has evaluated the effects on multiple levels, which has strengthened the article with powerful results.

 However, the introduction lacks detailed information on the metabolism pathway and scientific explanation. Also, it is recommended to use more recent references, not from 2016, when discussing the accurate confirmation of the stage and severity of NAFLD.

 « Challenges in addressing NAFLD include early 67 identification, as clinical indicators such as hepatic enzyme levels can be normal in pa- 68 tients with advanced disease. Neither serum nor radiological scanning can accurately con- 69 firm the stage and severity of the disease »

Regarding the body composition analysis using PXIMUS, it should be noted that the mice were anaesthetized by ketamine injection. However, several studies have shown the effect of the anaesthetic ketamine on changes in blood glucose and fatty acid levels.

Furthermore, the discussion section needs improvement in terms of explaining the metabolism pathway and providing scientific comparisons with previous studies. In many parts of the discussion, the author simply restates the results without a thorough scientific explanation!

The author must show the major revisions made in the text by highlighting the changes in a different colored text. 

It is imperative to consider all these remarks to reinforce the manuscript's quality and conclude more accurately.

Author Response

We appreciate that the reviewer recognize the significance of our study and provided us with helpful comments. We have carefully addressed all the points raised in our revised manuscript using the blue color. In addition, the authors revised sentences regarding grammatical errors throughout the manuscript.

Reviewer(s)' Comments to Author:

Reviewer 2:

  1. However, the introduction lacks detailed information on the metabolism pathway and scientific explanation. Also, it is recommended to use more recent references, not from 2016, when discussing the accurate confirmation of the stage and severity of NAFLD. “Challenges in addressing NAFLD include early identification, as clinical indicators such as hepatic enzyme levels can be normal in patients with advanced disease. Neither serum nor radiological scanning can accurately confirm the stage and severity of the disease”

Thank you for the comment. We have now provided simple background on insulin metabolism related to metabolic syndrome in the introduction (lines 80-91). Also, we updated recent references from 2021 to 2023, including reference 16-19 (lines 66-74 and 76-79).

  1. Regarding the body composition analysis using PXIMUS, it should be noted that the mice were anaesthetized by ketamine injection. However, several studies have shown the effect of the anesthetic ketamine on changes in blood glucose and fatty acid levels.

We thank the reviewer for suggestion. We agree that anesthetizing mice may alter blood glucose and fatty acid levels. In our study, PIXIMUS scans were conducted a day before necropsy, and blood glucose measurements were not taken after ketamine injection. Additionally, we provided a recovery period of more than 15-24 hours before necropsy, and all mice went through identical conditions. Therefore, the effect on biomarkers like hepatic triglycerides might not be affected by this.

  1. Furthermore, the discussion section needs improvement in terms of explaining the metabolism pathway and providing scientific comparisons with previous studies. In many parts of the discussion, the author simply restates the results without a thorough scientific explanation!

Thank you for your suggestion. We have added more scientific context related to metabolic disturbances and compared our research with previous findings. Thorough changes were made in the Discussion.

  1. The author must show the major revisions made in the text by highlighting the changes in a different colored text.

We have highlighted all the changes using blue color. 

Reviewer 3 Report

Comments and Suggestions for Authors

The work is very well designed, logically organized with the content of new information from the field of nutritional medicine.

1.       Is it true that there is a greater risk of NAFLD in females, since peak incidence occurs in higher age compared to males? – line 66

2.       5 mm slices of tissue are too thick for histological observation, do not you mean 5 µm? – line 186

3.       10x magnification – do you mean 10x objective or 10x total magn. (if you mean 10 objective x 10x eyepiece = 100x total) – line 189

Comments on the Quality of English Language

 Minor editing is required.

Author Response

We appreciate that the reviewer recognize the significance of our study and provided us with helpful comments. We have carefully addressed all the points raised in our revised manuscript using the blue color. In addition, the authors revised sentences regarding grammatical errors throughout the manuscript.

Reviewer(s)' Comments to Author:

Reviewer 3:

  1. Is it true that there is a greater risk of NAFLD in females, since peak incidence occurs in higher age compared to males? – line 66

Although the overall incidence of NAFLD is higher in males compared to females, the incidence of NAFLD in females with menopause, making age-associated NAFLD in females concerning. We clarified the statement clearer by adding, “Additionally, it is noteworthy that NAFLD prevalence is higher in men compared to women [10], with variations observed post-menopause [11-15]” in lines 66-68.

  1. 5 mm slices of tissue are too thick for histological observation; do not you mean 5 µm? – line 186

Thank you very much for bringing this to our attention. The section obtained was 5 µm, and we corrected that in our manuscript (line 205).

  1. 10x magnification – do you mean 10x objective or 10x total magn. (if you mean 10 objective x 10x eyepiece = 100x total) – line 189

The system used have a single lens, and we used a total magnification of 10X for taking pictures of the section. This information has updated in the manuscript (line 208). 

  1. Minor editing is required.

We have reviewed the entire manuscript for any errors in English.

Round 2

Reviewer 2 Report

Comments and Suggestions for Authors

In this version of the article “Curcumin-Rich Diet Mitigates Non-Alcoholic Fatty Liver Disease (NAFLD) by Attenuating Fat accumulation and Improving Insulin Sensitivity in Aged Female Mice Under Nutritional Stress” We can see an acceptable evolution compared to the first version because it has become more structured with more explanation.

the authors have relatively taken the reviewer's remarks and suggestions into consideration, which has positively impacted the quality and consistency of the article.

with this version, the article shows a good scientific level and represents an added value in the research topics that are interested in curcumin-rich diet beneficial effects

the article is accepted for me with this version

Author Response

In this version of the article “Curcumin-Rich Diet Mitigates Non-Alcoholic Fatty Liver Disease (NAFLD) by Attenuating Fat accumulation and Improving Insulin Sensitivity in Aged Female Mice Under Nutritional Stress” We can see an acceptable evolution compared to the first version because it has become more structured with more explanation.

The authors have relatively taken the reviewer's remarks and suggestions into consideration, which has positively impacted the quality and consistency of the article.

With this version, the article shows a good scientific level and represents an added value in the research topics that are interested in curcumin-rich diet beneficial effects.

The article is accepted for me with this version.

<Response>

We appreciate your constructive comments and thoughtful consideration.